# Scaling Equipment Effect on Technical–Tactical Actions in U-13 Basketball Players: A Maturity Study

Enrique Ortega-Toro [1,2,3], Ricardo André Birrento-Aguiar [1,2,3,*], José María Giménez-Egido [1,2,3,4], Francisco Alarcón-López [3,4] and Gema Torres-Luque [2,3,5]

1 Faculty of Sports Sciences, University of Murcia, 30720 Santiago de la Ribera, Spain; eortega@um.es (E.O.-T.); josemaria.gimenez@um.es (J.M.G.-E.)
2 Human Movement and Sports Science, HUMSE, Faculty of Sports Sciences, University of Murcia, 30720 Santiago de la Ribera, Spain; gtluque@ujaen.es
3 Sports Performance Analysis Association (SPAA), 30720 Santiago de la Ribera, Spain; f.alarcon@ua.es
4 Faculty of Education, University of Alicante, 03690 Sant Vicent del Raspeig, Spain
5 Department of Plastic, Music and Corporal Expression, Faculty of Humanities and Education Science, University of Jaén, 23071 Jaén, Spain
* Correspondence: ra.birrentoaguiar@um.es; Tel.: +34-630-00-35-79

**Abstract:** The aim of this study was to analyse the performance of technical–tactical actions in two different types of tournaments and the influence of biological age on the performance of young basketball players. Thirty-seven under-13 male basketball players (age = 12.91 ± 0.57 years) were selected from four southeast Spanish teams to participate in two different tournaments on two consecutive days. The following technical–tactical variables were analysed: (a) Ball Obtained; (b) Ball Handler Player Actions; (c) Ball Handler Player Finished Actions; and (d) Ball Handler Shooting Performance. The results showed that reduced basket height and a closer three-point line promoted a higher number of balls obtained, 1 vs. 1 situations, finished ball player actions, shots, and the efficacy of offence phases. There was a significant increase in the number of balls obtained, 1 vs. 1 situations played, the number of plays finished with a lay-up or shot, number of received personal fouls, number of plays finished in 1 vs. 2, and those finished in equality and inferiority with a high defence opposition. The modified version presented a higher number of technical–tactical actions in Late Maturity players. The authors of this study believe that it is necessary to conduct more experimental studies and use bio-banding strategies in young basketball competitions.

**Keywords:** early; late; maturity; player actions; young

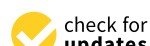


## 1. Introduction

Basketball is a late specialization sport [1]. To reach the end of this process, the player ought to go through different stages of the training process. Replicating the training and competitive processes of professional basketball is often common in young teams [2]. In these cases, the maturational development of the child and adolescent is not used in young competition systems. Competitions at the initial level copy the adult basketball model, without considering the physical and cognitive characteristics of young players [3–6]. A relevant fact is the team's separation by chronological age and not by biological age, thus creating an unequal and unmotivating competition. However, there is evidence to show that a biological age classification benefits their performance, as well as the teaching and learning process [7–12]. In addition, there are studies that show that an adapted-rules competition brings benefits to the development of different aspects, such as increasing the number of shots, increasing the efficiency in shooting, increasing the number of passes, or achieving the highest levels of self-efficacy [10,13–17]. In particular, research studies have been conducted in which the basket height has been reduced, and an increase in single actions (shooting and 1 vs. 1 actions) and team game situations (fast breaks and offence positional

phases) has been found [17,18]. On the other hand, another study manipulated the size of the ball by pointing out that it improves the shot accuracy and efficacy, specifically on free throws and three-point shots [19,20]. All these studies used modified rules; however, they did not offer proposals for a specific competition response. The change from minibasket to U-13 level means that the rules were adapted to the adult rules at a very early stage in the players' development. According to the above-mentioned evidence, an intermediate step between adapted basketball and adult basketball should be sought. Therefore, this study analyses and compares the influence of a modified competition on technical–tactical actions in U-13 players.

## 2. Materials and Methods

### 2.1. Participants

In this study, four teams participated in two different tournaments on two following days. Thirty-seven under-13 male basketball players (age = 12.91 ± 0.57 years) participated. All players committed themselves to the research group, participated in the pre-tournament data collection, and performed all the proposed activities during the formal tournament and the modified tournaments. Written informed consent was obtained from all participants and their parents before the research. This study was approved by the Institutional Research Ethics Committee of the University of Murcia.

### 2.2. Procedures

Each team played 3 matches per tournament. In both tournaments, each team played against each other; therefore, in both tournaments, each team played the same matches, against the same opponents in the same order, and with the same players per quarter. On the first day, the tournament was played according to the rules of the Spanish Basketball Federation (FEB) for under-14 players. Indeed, the rules for the first, second, and third quarters do not change. All players registered on the scoresheet play a minimum of one quarter played in the first 3 quarters and a maximum of 2 quarters played in a row. No changes were allowed, except for injuries or exclusions for 5 fouls. On the second day (Modified Tournament), the height of the basket was lowered to 2.90 m from the basket (Figure 1). All shots taken behind this line were valued at 3 points, and all shots made from the 6.75 m line were valued at 4 points. Before the tournament, somatic maturation data were collected. A non-invasive method appropriated for the age range of the participants was used, considering anthropometric data (body mass, standing height, leg length, and sitting height) and chronological age. The height of each player's mother and father was calculated to determine the APHV (age at peak height velocity) of each player.

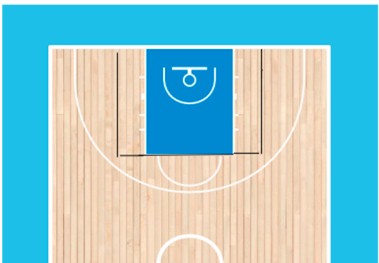

**Figure 1.** Modified Tournament half court.

Somatic Maturation

Height was recorded using a commercially portable stadiometer (Tanita BF-522W, Tokyo, Japan, nearest 0.1 cm). Body mass was estimated using a scale (Tanita BF-522W, Tokyo, Japan, nearest 0.1 kg). All measurements were taken following the guidelines outlined by the International Society for the Advancement of Kinanthropometry (ISAK) by the same researcher, who holds an ISAK Level 1 accreditation. Players' height, weight, chronological age, and mid-parent height were used to predict the adult height of each

player [21]. The height of the biological parents of each player was self-reported and adjusted for overestimation using previously established equations [22]. The current height of each player was then expressed as a percentage of their predicted adult height (% PAH), which can then be used as an index of somatic maturation [23]. Players were grouped into two maturity timing bands based on z-scores: average, on-time to late (z-score between +0.5 and < −0.5), and early (z-score > +0.5).

### 2.3. Variables

The independent variable was the game format. There were two different types of games: FEB official rules and Modified Rules The dependent variables were (a) Ball Obtained (Ball Obtained, Static Ball Obtained, Dynamic Ball Obtained, Pass Reaches, Pass Not Reaches, Pass Out-Out, Pass Out-In, Pass In-In, Pass In-Out and Out Court Pass); (b) Ball Handler Player Actions (NOT Play 1 vs.1, Play 1 vs. 1, 1 vs. 1 Not Outperform the Opponent, NOT Play TMCB and Play TMCB); (c) Ball Handler Player Finished Actions (Received Personal Foul, Total Turn-Overs, Pass, Finished on 1 vs. 0, Finished on 1 vs. 1, Finished on 2 vs. 2, Finished on 1 vs. 2, Finished on Numerical Superiority, Finished on Numerical Superiority, Finished on Numerical Equality, Finished on Numerical Inferiority, High Opposition, Medium Opposition, Low Opposition, and Minimum Opposition) and (d) Ball Handler Shooting Performance (Lay-up, Jump Shot, Total Shots, 2 Points Shot, 3 Points Shot, 4 Points Shot, Shooting in Game (%), Performance: 0 points, Performance: 1 point, Performance: 2 points, Performance: 3 points, Performance: 4 points, Not Effective Offence Phase, Effective Offence Phase, and Effective Offence Phase, (%)).

Observational methodology was used to record the data [24]. For data quality control, an observer training proposal was developed [25]. Training of two observers (students of the degree of physical activity and sport sciences) was conducted. Minimum reliability values (0.95) were obtained.

### 2.4. Data Analysis

Data are presented as mean $\pm$ SD. Normality of data distribution and homoscedasticity were confirmed using the Shapiro–Wilk statistic and Levene's test for equality of variances; thus, parametric analyses were used. The related samples t-test was used to analyse within-group changes. A 2 vs. 2 mixed-model analysis of variance (ANOVA) was performed on the absolute values of all parameters to determine the main effects between maturity timing groups and competition models. Effect sizes were evaluated using an omega squared ($\omega^2$), with <0.06, 0.06–0.14, and >0.14 indicating small, medium, and large effects, respectively. The sample sizes were evaluated using a power of 0.80, alpha = 0.05, and a medium effect size (f = 0.25). All statistical analyses were performed using JASP software (version 0.13, University of Amsterdam, Amsterdam, The Netherlands) and G Power 3.1.9.7.

### 3. Results

#### 3.1. Ball Obtained

In general, Table 1 shows that, in the Modified Tournament, a lower number in the following variables was found: (a) Out-In Passes; (b) In-In Passes; (c) In-Out Passes, and (d) Out Court Passes. No significant differences were observed.

When analysing all the players in the Modified Tournament, there were a greater number of actions in the variables related to obtaining the ball, with significant differences in (a) Total Ball Obtaining (Z = −2.138, *p* = 0.033) and (b) Out-Out pass (Z = −1.965, *p* = 0.049).

When analysing each of the groups of players, in the Modified Tournament, there were a great number of actions in the variables related to obtaining the ball, with significant differences in the Total Ball Obtaining, Dynamic Ball Obtained, Pass Reaches, Pass Out-Out, and Pass Out-In variables in the Late Maturity players. Lower significant values were found in the Static Ball Obtained variable. There were no significant differences in any variable analysed between the tournaments in the Early Maturity players.

**Table 1.** Mean values and standard deviation of Ball Obtained variables, according to maturating timing and tournament.

| Variables | Early (n = 25) | | Late (n = 12) | | Total (n = 37) | |
|---|---|---|---|---|---|---|
| | FEB | Modified | FEB | Modified | FEB | Modified |
| Ball Obtained | 99.16 ± 48.28 | 95.64 ± 40.3 | 79.08 ± 32.46 | 97.58 ± 30.98 | 92.65 ± 44.35 | 96.27 ± 37.11 * |
| Static Ball Obtained | 9.40 ± 4.58 | 13.32 ± 8.54 | 13.83 ± 4.15 | 8.58 ± 4.42 | 10.84 ± 4.87 | 11.78 ± 7.72 |
| Dynamic Ball Obtained | 89.76 ± 47.04 | 82.32 ± 35.26 | 65.25 ± 34.38 | 89.00 ± 30.31 | 81.81 ± 44.4 | 84.49 ± 33.46 |
| Pass Reaches | 8.16 ± 6.3 | 7.80 ± 4.99 | 4.75 ± 2.99 | 6.75 ± 3.65 | 7.05 ± 5.64 | 7.46 ± 4.57 |
| Pass Not Reaches | 66.76 ± 33.51 | 68.24 ± 32.9 | 54.83 ± 25.18 | 56.58 ± 35.36 | 62.89 ± 31.22 | 64.46 ± 33.76 |
| Pass Out-Out | 59.32 ± 31.79 | 62.84 ± 32.48 | 45.67 ± 22.21 | 58.00 ± 30.63 | 54.89 ± 29.43 | 61.27 ± 31.55 * |
| Pass Out-In | 5.64 ± 3.55 | 4.92 ± 3.28 | 4.75 ± 4.05 | 6.17 ± 4.15 | 5.35 ± 3.68 | 5.32 ± 3.58 |
| Pass In-In | 2.56 ± 2.8 | 2.08 ± 1.93 | 2.83 ± 2.55 | 3.17 ± 4.22 | 2.65 ± 2.69 | 2.43 ± 2.86 |
| Pass In-Out | 5.64 ± 3.72 | 4.76 ± 3.95 | 4.42 ± 3.12 | 4.17 ± 2.92 | 5.24 ± 3.54 | 4.57 ± 3.62 |
| Out Court Pass | 0.68 ± 1.18 | 0.52 ± 0.71 | 0.83 ± 0.94 | 0.75 ± 0.97 | 0.73 ± 1.1 | 0.59 ± 0.8 |

* Statistical significance $p < 0.05$.

The variance of two factors analysis (2 vs. 2), maturation level (Early Maturity vs. Late Maturity) and tournament (FEB Tournament vs. Modified Tournament) were analysed. From analysing the last factor repeated measures, it can be seen that the interaction effect of the tournament factor by maturation level is not significant in the following variables: (a) Pass Reach ($F_{1,35}$ = 2.318, $p$ = 0.137, $\eta^2$ = 0.062); (b) Pass Not Reach ($F_{1,35}$ = 0.001, $p$ = 0.979, $\eta^2$ = 0.000); (c) Out-Out Pass ($F_{1,35}$ = 1.131, $p$ = 0.295, $\eta^2$ = 0.031); (d) In-In Pass ($F_{1,35}$ = 0.835, $p$ = 0.367, $\eta^2$ = 0.023); (e) In-Out Pass ($F_{1,35}$ = 0.145, $p$ = 0.706, $\eta^2$ = 0.004), and (f) Out Court Pass ($F_{1,35}$ = 0.031, $p$ = 0.862, $\eta^2$ = 0.001).

On the other hand, an interaction effect is observed in the tournament factor by maturity level. This is significant in the following variables: (a) Static Ball Obtained ($F_{1,35}$ = 14.895, $p$ = 0.000, $\eta^2$ = 0.299); (b) Dynamic Ball Obtained ($F_{1,35}$ = 7.193, $p$ = 0.011, $\eta^2$ = 0.170); and (c) Out-In Pass ($F_{1,35}$ = 4.595, $p$ = 0.039, $\eta^2$ = 0.116). A tendency to significance was observed in the Ball Obtained variable ($F_{1,35}$ = 3.495, $p$ = 0.070, $\eta^2$ = 0.091).

In this sense, the Obtained Ball variable (Figure 2) increased in the Modified Tournament. Specifically, this was observed in the players with late development and a slight decrease in those with early development. However, a tendency to significance was observed only in the players with late development ($p$ = 0.064).

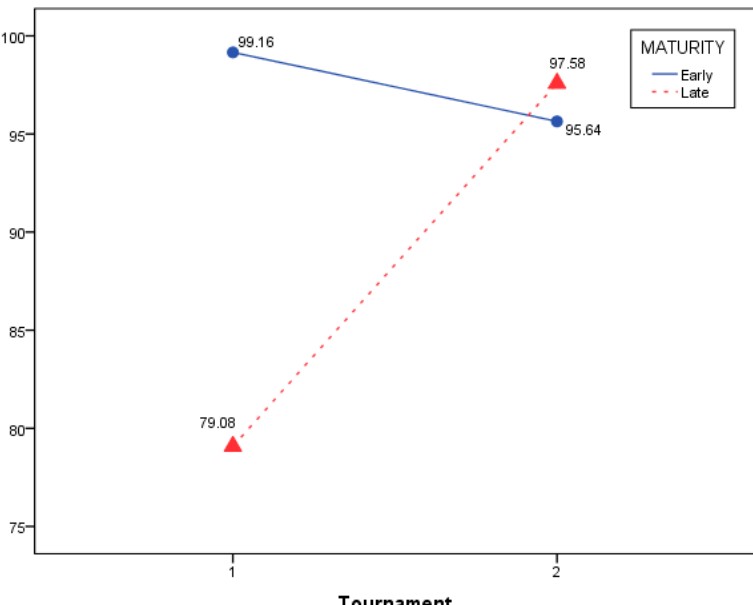

**Figure 2.** Ball Obtained evolution, according to maturating timing and tournament.

In the Static Ball Obtained variable (Figure 3), a significant increase was observed for the Modified Tournament and Early Maturity players ($p = 0.008$), and a significant decreased tendency was observed for the Late Maturity players ($p = 0.081$).

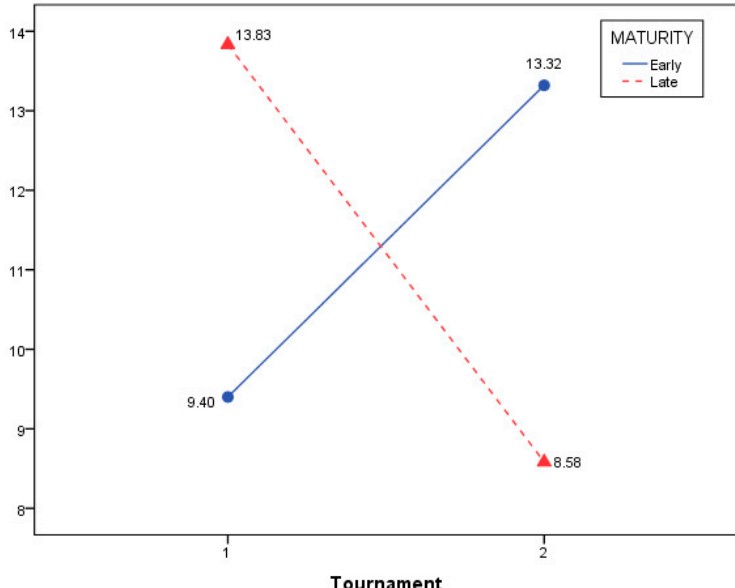

**Figure 3.** Static Ball Obtained evolution, according to maturating timing and tournament.

On the other hand, in the Dynamic Ball Obtained variable (Figure 4), there was a significant increase in the Modified Tournament in the Late Maturity players ($p = 0.018$), and a slight decrease that was not significant in the Early Maturity players.

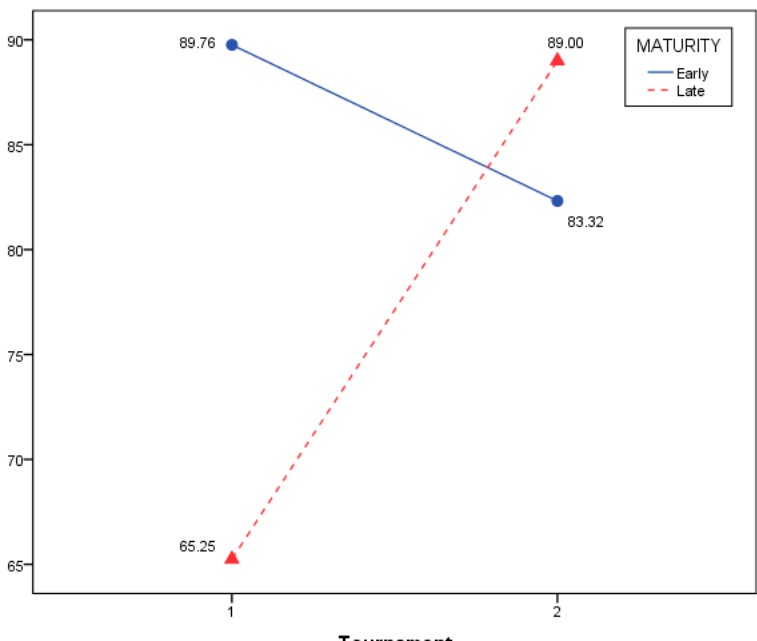

**Figure 4.** Dynamic Ball Obtained evolution, according to maturating timing and tournament.

Finally, there was a significant increase in the number of Out-In Passes by Late Maturity players ($p = 0.093$), and a slight non-significant decrease in the Early Maturity players (see Figure 5).

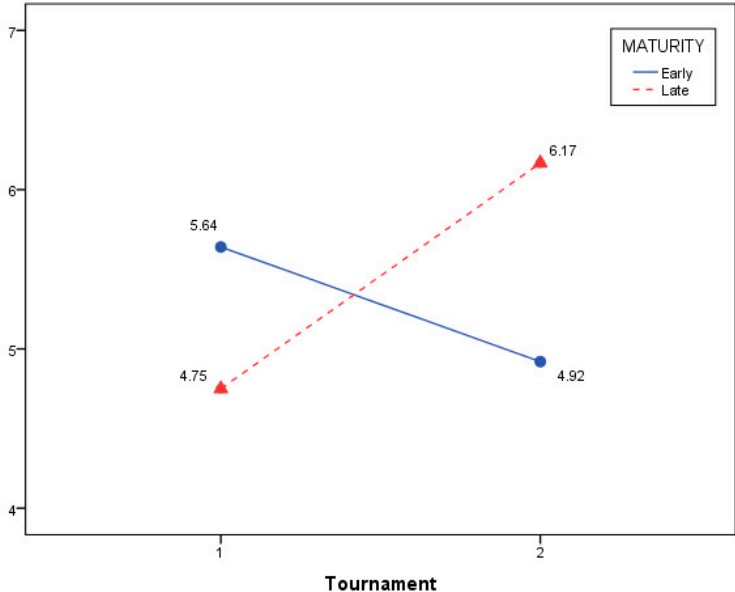

**Figure 5.** Out-In Passes evolution, according to maturating timing and tournament.

### 3.2. Ball Handler Player Actions

When analysing all the players in the Modified Tournament, Table 2 shows a lower number of 1 vs. 1 actions where the offence player does not overcome the 1 vs. 1. A lower number of actions with TMCB were observed. Only in the 1 vs. 1 NOT Outperform the Opponent variable were significant differences observed ($Z = -2.570$, $p = 0.010$).

**Table 2.** Mean values and standard deviation of Ball Handler Player Actions, according to maturating timing and tournament.

| Variables | Early (n = 25) | | Late (n = 12) | | Total (n = 37) | |
| --- | --- | --- | --- | --- | --- | --- |
| | FEB | Modified | FEB | Modified | FEB | Modified |
| NOT Play 1 vs. 1 | $98.44 \pm 46.2$ | $103.20 \pm 40.57$ | $73.67 \pm 29.41$ | $88.33 \pm 48.59$ | $90.41 \pm 42.72$ | $98.38 \pm 43.22$ |
| Play 1 vs. 1 | $12.68 \pm 9.79$ | $12.86 \pm 6.47$ | $14.08 \pm 8.68$ | $17.75 \pm 5.51$ | $13.14 \pm 9.35$ | $13.67 \pm 6.77$ |
| 1 vs. 1 Outperform the Opponent | $8.40 \pm 8.24$ | $9.80 \pm 5.71$ | $8.45 \pm 5.84$ | $11.92 \pm 3.68$ | $8.42 \pm 7.5$ | $10.01 \pm 5.3$ |
| 1 vs. 1 NOT Outperform the Opponent | $4.16 \pm 3.31$ | $3.06 \pm 1.61$ | $6.91 \pm 3.86$ | $5.83 \pm 2.44$ | $5.00 \pm 3.66$ | $3.66 \pm 2.39$ * |
| NOT Play TMCB | $104.00 \pm 46.85$ | $105.44 \pm 43.14$ | $80.00 \pm 32.65$ | $91.58 \pm 50.84$ | $96.22 \pm 43.8$ | $100.95 \pm 45.54$ |
| Play TMCB | $7.40 \pm 7.17$ | $7.00 \pm 7.39$ | $10.50 \pm 5.62$ | $7.75 \pm 6.4$ | $8.41 \pm 6.79$ | $7.24 \pm 7.01$ |

* Statistical significance $p < 0.05$.

When analysing each of the groups of players in the Modified Tournament, there was a great number of actions in the variables related to Ball Obtained, with significant differences in the Not Play 1 vs. 1 and Play 1 vs. 1 in the Late Maturity players. There were no significant differences in any variable analysed between tournaments in the Early Maturity players.

The variance of two factors analysis (2 vs. 2), maturation level (Early Maturity vs. Late Maturity) and tournament (FEB Tournament vs. Modified Tournament) were analysed. Through analysing the repeated measures in the last factor, it can be seen that the interaction effect of the tournament factor by maturation level is not significant in the following variables: (a) Does not exist $1 \times 1$ ($F_{1,35} = 0.380$, $p = 0.542$, $\eta^2 = 0.011$); (b) Does not surpass the opponent in the $1 \times 1$ ($F_{1,35} = 0.028$, $p = 0.868$, $\eta^2 = 0.001$); (c) Not Play TMCB ($F_{1,35} = 0.378$, $p = 0.543$, $\eta^2 = 0.011$); (d) Play TMCB ($F_{1,35} = 1.198$, $p = 0.281$, $\eta^2 = 0.033$).

### 3.3. Ball Handler Player Finished Actions

When analysing all the players in the Modified Tournament, Table 3 shows a lower number in the following variables: (a) 2 vs. 2; (b) Numerical Superiority; (c) Medium Opposition, and (d) Low Opposition. Only in the Low Opposition variable were there significant differences observed ($Z = -2.108$, $p = 0.035$). In the Modified Tournament, there were a greater number of actions in the rest of the variables related to the Player Finished Actions. Significant differences were observed in the following variables: (a) 1 vs. 2 ($Z = -2.138$, $p = 0.033$), (b) Numerical Inferiority ($Z = -2.385$, $p = 0.017$), (c) Received Personal Foul ($Z = -1.998$, $p = 0.042$), and (d) High Opposition ($Z = -2.141$, $p = 0.038$).

**Table 3.** Mean values and standard deviation of Ball Handler Player Finished Actions, according to maturing timing and tournament.

| Variables | Early (n = 25) | | Late (n = 12) | | Total (n = 37) | |
|---|---|---|---|---|---|---|
| | **FEB** | **Modified** | **FEB** | **Modified** | **FEB** | **Modified** |
| Received Personal Foul | $2.76 \pm 1.61$ | $3.64 \pm 2.9$ | $3.17 \pm 2.66$ | $4.25 \pm 3.44$ | $2.89 \pm 2.61$ | $3.84 \pm 3.16$ * |
| Total Turn-Overs | $10.68 \pm 6.92$ | $10.20 \pm 5.70$ | $9.00 \pm 4.09$ | $10.75 \pm 6.21$ | $10.14 \pm 6.14$ | $10.38 \pm 5.79$ |
| Pass | $67.84 \pm 33.9$ | $69.44 \pm 33.8$ | $55.00 \pm 25.1$ | $58.00 \pm 34.5$ | $63.68 \pm 31.56$ | $65.72 \pm 33.98$ |
| 1 vs. 0 | $20.28 \pm 9.86$ | $20.80 \pm 11.38$ | $13.58 \pm 10.18$ | $14.83 \pm 14.1$ | $18.11 \pm 10.33$ | $18.86 \pm 12.45$ |
| 1 vs. 1 | $67.44 \pm 63.42$ | $66.00 \pm 26.13$ | $56.92 \pm 21.09$ | $66.00 \pm 26.13$ | $64.03 \pm 32.33$ | $66.00 \pm 3.13$ |
| 2 vs. 2 | $9.24 \pm 5.13$ | $8.40 \pm 4.37$ | $6.75 \pm 3.86$ | $7.25 \pm 5.88$ | $8.43 \pm 4.85$ | $8.03 \pm 4.86$ |
| 1 vs. 2 | $8.12 \pm 6.43$ | $11.76 \pm 7.63$ | $8.75 \pm 5.5$ | $10.17 \pm 7.30$ | $8.32 \pm 6.07$ | $11.24 \pm 7.46$ * |
| Numerical Superiority | $24.72 \pm 11.82$ | $23.44 \pm 12.21$ | $16.58 \pm 10.01$ | $16.67 \pm 15.13$ | $22.08 \pm 11.78$ | $21.24 \pm 13.41$ |
| Numerical Equality | $76.68 \pm 39.05$ | $74.40 \pm 37.2$ | $63.67 \pm 21.97$ | $69.75 \pm 33.11$ | $72.46 \pm 34.67$ | $72.89 \pm 35.53$ |
| Numerical Inferiority | $9.52 \pm 7.81$ | $13.96 \pm 8.85$ | $10.25 \pm 6.44$ | $12.25 \pm 9.11$ | $9.76 \pm 7.31$ | $13.41 \pm 8.81$ * |
| High Opposition | $39.76 \pm 22.91$ | $44.56 \pm 21.07$ | $34.33 \pm 10.88$ | $41.92 \pm 23.57$ | $38.00 \pm 19.82$ | $43.70 \pm 21.62$ * |
| Medium Opposition | $36.92 \pm 19.33$ | $34.04 \pm 20.08$ | $28.83 \pm 11.38$ | $30.33 \pm 12.94$ | $34.30 \pm 17.42$ | $32.84 \pm 17.97$ |
| Low Opposition | $9.60 \pm 6.49$ | $7.40 \pm 6.1$ | $9.92 \pm 6.73$ | $8.17 \pm 5.29$ | $9.70 \pm 6.48$ * | $7.65 \pm 5.78$ |
| Minimum Opposition | $24.84 \pm 12.87$ | $26.08 \pm 12.98$ | $16.92 \pm 10.94$ | $18.83 \pm 16.94$ | $22.27 \pm 12.69$ | $23.73 \pm 14.55$ |

* Statistical significance $p < 0.05$.

When analysing each of the groups of players in the Modified Tournament, there were a great number of actions in variables related to Fouls Received, Total Turnovers, Pass, 1 vs. 1, 1 vs. 2, Numerical Equality, Numerical Inferiority and High Opposition in the Late Maturity players. In addition, in the Modified Tournament, there were a great number of actions in the variables related to 1 vs. 2 and Numerical Inferiority in the Early Maturity players.

The variance of two factors analysis (2 vs. 2), maturation level (Early Maturity vs. Late Maturity), and tournament (FEB Tournament vs. Modified Tournament) were analysed. Through analysing the repeated measures in the last factor, it can be seen that the interaction effect of the tournament factor by maturation level is not significant in the following variables: (a) Received Personal Foul ($F_{1,35} = 0.026$, $p = 0.873$, $\eta^2 = 0.001$); (b) Total Turnovers ($F_{1,35} = 1.138$, $p = 0.293$, $\eta^2 = 0.031$); (c) Pass ($F_{1,35} = 0.018$, $p = 0.893$, $\eta^2 = 0.001$); (d) 1 vs. 0 ($F_{1,35} = 0.027$, $p = 0.871$, $\eta^2 = 0.001$); e: 1 vs. 1 ($F_{1,35} = 1.361$, $p = 0.251$, $\eta^2 = 0.037$); (f) 2 vs. 2 ($F_{1,35} = 0.445$, $p = 0.509$, $\eta^2 = 0.013$); (g) 1 vs. 2 ($F_{1,35} = 0.476$, $p = 0.495$, $\eta^2 = 0.013$); (h) Received Superiority ($F_{1,35} = 0.075$, $p = 0.786$, $\eta^2 = 0.002$); (i) Received Equality ($F_{1,35} = 0.614$, $p = 0.439$, $\eta^2 = 0.017$); (j) Received Inferiority ($F_{1,35} = 0.457$, $p = 0.503$, $\eta^2 = 0.013$); (k) High Opposition ($F_{1,35} = 0.118$, $p = 0.733$, $\eta^2 = 0.003$); (l) Medium Opposition ($F_{1,35} = 0.977$, $p = 0.330$, $\eta^2 = 0.027$); (m) Low Opposition ($F_{1,35} = 0.055$, $p = 0.815$, $\eta^2 = 0.002$), and (n) Minimum Opposition ($F_{1,35} = 0.016$, $p = 0.900$, $\eta^2 = 0.000$).

### 3.4. Ball Handler Shooting Performance

Through analysing all the players in the Modified Tournament, Table 4 shows a lower number of following variables: (a) Lay-up, (b) 2-point shot and (c) performance 1-point. No significant differences were observed. On Modified Tournament, there were a greater number of actions in the rest of the variables related to the actions related to the Ball Handler Shooting Performance. Significant differences were observed on follow variables: (a) Jump Shot (Z = −2.594, *p* = 0.009), (b) Effective Offence Phase (Z = −3.129, *p* = 0.002) and (c) Effective Offence Phase (%) (Z = −2.325, *p* = 0.020). Trends towards significance were observed on (a) Total Shots (Z = −1.722, *p* = 0.085), and (b) Shooting in Game (%) (Z = −1.712, *p* = 0.087).

**Table 4.** Mean values and standard deviation of Ball Handler Shooting Performance, according to maturating timing and tournament.

| Variables | Early (n = 25) | | Late (n = 12) | | Total (n = 37) | |
|---|---|---|---|---|---|---|
| | FEB | Modified | FEB | Modified | FEB | Modified |
| Lay-up | 10.84 ± 9.34 | 9.72 ± 6.64 | 9.67 ± 7.5 | 9.92 ± 5.3 | 10.46 ± 8.7 | 9.78 ± 6.16 |
| Jump Shot | 13.88 ± 6.52 | 16.16 ± 7.05 | 9.17 ± 5.27 | 12.67 ± 8.26 | 12.35 ± 6.47 | 15.03 ± 7.53 * |
| Total Shots | 24.72 ± 13.33 | 25.88 ± 11.88 | 18.83 ± 9.39 | 22.58 ± 12.72 | 22.81 ± 12.38 | 24.81 ± 12.08 |
| 2 Points Shot | 20.60 ± 12.59 | 17.28 ± 10.09 | 16.25 ± 8.61 | 15.08 ± 8.6 | 19.19 ± 11.52 | 16.57 ± 9.57 |
| 3 Points Shot | 4.12 ± 3.33 | 3.68 ± 3.67 | 2.58 ± 3.37 | 3.25 ± 2.96 | 3.62 ± 3.38 | 3.54 ± 3.42 |
| 4 Points Shot | - | 4.92 ± 4.28 | - | 4.25 ± 3.33 | - | 4.70 ± 3.96 |
| Shooting in Game (%) | 33.16 ± 11.78 | 40.19 ± 12.39 | 38.13 ± 13.28 | 36.45 ± 14.47 | 34.77 ± 12.33 | 38.98 ± 13.02 |
| Performance: 0 Points | 16.24 ± 8.4 | 15.44 ± 7.43 | 11.08 ± 4.42 | 14.25 ± 8.04 | 14.57 ± 7.68 | 15.05 ± 7.54 |
| Performance: 1 Point | 0.88 ± 1.27 | 0.56 ± 0.96 | 0.75 ± 0.75 | 1.17 ± 1.9 | 0.84 ± 1.12 | 0.76 ± 1.34 |
| Performance: 2 Points | 7.64 ± 5.72 | 8.52 ± 5.65 | 6.92 ± 6.24 | 6.25 ± 4.92 | 7.41 ± 5.82 | 7.78 ± 5.46 |
| Performance: 3 Points | 0.84 ± 1.11 | 1.00 ± 1.32 | 0.83 ± 1.19 | 1.50 ± 1.24 | 0.84 ± 1.12 | 1.16 ± 1.3 |
| Performance: 4 Points | - | 0.92 ± 1.04 | - | 0.58 ± 1.24 | - | 0.81 ± 1.1 |
| Not Effective Offence Phase | 15.56 ± 6.62 | 16.28 ± 7.23 | 12.58 ± 35.54 | 13.83 ± 7.63 | 14.59 ± 6.98 | 15.49 ± 7.34 |
| Effective Offence Phase | 8.96 ± 6.19 | 11.08 ± 7.44 | 6.67 ± 4.19 | 10.00 ± 7.35 | 8.22 ± 5.67 | 10.73 ± 7.32 * |
| Effective Offence Phase (%) | 34.57 ± 12.89 | 38.51 ± 12.1 | 35.54 ± 11.51 | 41.21 ± 13.96 | 34.88 ± 12.31 | 39.39 ± 12.6 * |

* Statistical significance *p* < 0.05.

When analysing each of the groups of players in the Modified Tournament, there were a great number of actions in variables related to total shots, 3-point shots, 3-point performance, and effective offensive phases and effective offensive phases percentage in the Late Maturity players. In addition, in the Modified Tournament, there were a great number of actions in variables related to Jump Shot in Early Maturity players.

The variance of two factors analysis (2 vs. 2), maturation level (Early Maturity vs. Late Maturity) and tournament (FEB Tournament vs. Modified Tournament) were analysed. Analyzing repeated measures in the last factor, it can be seen that the interaction effect of the tournament factor by maturation level is not significant in the following variables: (a) Lay-up ($F_{1,35}$ = 0.238, *p* = 0.629, $\eta^2$ = 0.007); (b) Jumps Shot ($F_{1,35}$ = 0.222, *p* = 0.641, $\eta^2$ = 0.006); (c) Total Shots ($F_{1,35}$ = 0.318, *p* = 0.577, $\eta^2$ = 0.009); (d) 2 Points Shot ($F_{1,35}$ = 0.287, *p* = 0.596, $\eta^2$ = 0.008); (e) 3 Points Shot ($F_{1,35}$ = 0.456, *p* = 0.504, $\eta^2$ = 0.013); (f) Shooting in Game (%) ($F_{1,35}$ = 3.492, *p* = 0.070, $\eta^2$ = 0.091); (g) Performance: 0 points ($F_{1,35}$ = 1.829, *p* = 0.185, $\eta^2$ = 0.050); (h) Performance: 1 point ($F_{1,35}$ = 1.704, *p* = 0.200, $\eta^2$ = 0.046); (i) Performance: 2 points ($F_{1,35}$ = 0.608, *p* = 0.441, $\eta^2$ = 0.017); (j) Performance: 3 points ($F_{1,35}$ = 0.639, *p* = 0.430, $\eta^2$ = 0.018); (k) Not Effective Offence Phase ($F_{1,35}$ = 0.053, *p* = 0.820, $\eta^2$ = 0.002); (l) Effective Offence Phase ($F_{1,35}$ = 0.018, *p* = 0.893, $\eta^2$ = 0.001); (m) Effective Offence Phase (%) ($F_{1,35}$ = 0.182, *p* = 0.672, $\eta^2$ = 0.005).

## 4. Discussion

Scientific evidence needs to be provided to sports federations in a concrete proposal to modify under-13 basketball. Specifically, the aim of this study was to analyse the lower basket height, reducing the distance of the three-point line, and adding a four-point line.

In the Modified Tournament, the players experienced different game situations typical of integral learning and the non-linear pedagogical approach [2,12,26]: (a) greater participation of the players in technical–tactical actions; (b) greater variability of technical–tactical actions; (c) more offensive than defensive actions; and (d) a greater efficiency of the finishing actions, despite a higher level of opposition.

Regarding the participation of the players in technical–tactical actions, the results showed a significant increase in the number of balls obtained in the modified tournament. This increase is related to a greater number of passes, possibly because, as there is a four-point line and the three-point line is closer to the basket, the defence players should be closer to the offence players, and this consequently increases defensive mismatches, to enable the player to find an effective place to shoot. The increase in the number of passes provokes the intervention of more players during the game, thus increasing participation and collective play [27]. On the one hand, the participation of a greater number of players per offence phase is a differential element between winners and losers [28,29]. In addition, the intervention of more players per offence phase promotes the participation of other players. This participation is one of the basic pillars in sport initiation [22].

Therefore, the proposed rule modifications are appropriate because they produce a greater number of passes per offence phase and promote the intervention of more players during the game.

The data from this study show that in the Modified Tournament, the players perform a greater variability of technical–tactical actions, both in the types of passes, types of shots, and 1 vs. 1 actions. This greater variability may be due not only to the existence of the three-point and four-point lines but also due to the basket height modification.

These rules mean that the defence players are closer to the offence players and it is necessary to carry out other technical–tactical actions to be able to overcome their opponents, with more feinting, exits, types of passes, etc. Furthermore, because of the modification of the height of the basket, the types of shots also vary in terms of the mode and place of the shot. In this sense, a multitude of studies have pointed out that one of the most important aspects of achieving an adequate teaching–learning process is to encourage variability and move away from early specialisation [30–34].

In this sense, it can be affirmed that the rule modification proposal is adequate, giving rise to a greater variability in the technical–tactical actions carried out by the players.

However, these data reflect a greater number of offensive actions than defensive ones and a greater shooting efficiency. In the Modified Tournament, players received more fouls, finished in 1 vs. 2 situations, and attempted to finish actions with greater defensive pressure. In other words, the offence players took more risks.

These results are probably due to the change in the height of the basket and the fact that players feel more stimulated to solve a complex tactical situation because the basket is more accessible. Similarly, these results could be due to the fact that in order to avoid shots by unopposed players from intermediate positions but whose value is three points (and even four points), defence players are forced to be closer to the ball player at large and middle distances, leaving inside zone positions and generating spaces close to the basket that encourage and facilitate 1 vs. 1 and 1 vs. 2 actions. This, in turn, provokes defensive help actions, generating advantages in outside positions and ball-passing games [35,36]. Similarly, it should be noted that higher effectiveness percentages were obtained; however, the opposition defence pressure when a player took a shot was higher. There were game situations where better percentages of effectiveness were obtained in the shot [37], meaning that the young players were making shots with an opposition defence pressure. These situations will be common in future basketball levels.

The players in this study made shots during the Modified Tournament in the lower-height basket. This situation is unusual for them, but they obtained better efficiency percentages; therefore, the strength factor is not so decisive [26,38]. If these players practiced with a modified basket height (2.90 m), their efficiency percentages would probably increase.

This improvement in shooting efficiency is probably due to the low height of the basket, which allows for the maintenance of a technical shooting pattern suitable for the physical development of the players. A study pointed out that modifications established in the transition from minibasket (2.50 m) to basketball (3.05 m) affect the technical shooting pattern [39,40]. This intermediate step seems to be the key to achieving a progressive teaching–learning process, with many basketball coaches and basketball researchers [41] pointing out the need for a progressive process in terms of the height of the basket.

Therefore, it can be affirmed that the proposed rule modifications are very appropriate as they cause greater variability in the type of shoots and shooting area, as well as more realistic throwing, with high opposition levels and better effectiveness percentages. Thus, in general, the proposed rule modifications lead to better game dynamics and more suitable environments to achieve an adequate teaching–learning process. There are a great number of offence phases, with more passes, a greater number of players participating in each of them, greater variability in technical–tactical actions, greater number of offensive actions, more shooting actions, and greater variability and effectiveness in these shots.

In addition to these modifications affecting all players, the results of the present study reflect that rule manipulation has a direct impact on Late Maturity players, minimising the physical difference between these players and Early Maturity players. The Modified Tournament allowed Late Maturity players to display greater participation, more variability, and greater efficiency during the game.

Analysing the results by maturity level, the Early Maturity players maintained similar values in both tournaments. The Late Maturity players increased their performance in the following variables: Ball Obtained, Dynamic Ball Obtained, Pass Reaches, Pass Out-Out, Pass Out-In, 1 vs. 1 play, 1 vs. 1 outplay, received personal fouls, Total Turnovers, Passing, 1 vs. 1, 1 vs. 2, Numerical Equality, Numerical Inferiority, High Opposition, Total Shots, 3-point shots, 3-point performance, effective offensive phases and effective and offensive phases percentage.

By analysing the Early Maturity player results, similar values in both tournaments were found. However, significant values were observed in the 1 vs. 2, Numerical Inferiority, Jump Shot, and Low Opposition variables in the Modified Tournament.

Analysing the maturity interaction effect factor, significant effects were found in the following variables: Static Ball Obtained Ball, Dynamic Ball Obtained, and Out-In Pass.

Many studies have indicated the need to control the maturation effect on appropriate competitions [7,41]. In most cases, it is necessary to create competitions between maturational levels and not by chronological age [22], but in many cases, this is not possible, so rule modifications could minimise this maturational effect. The combination of both aspects will be the key to a formative competition, generating competitive experiences that lead to improvements in the teaching–learning process for both types of athletes.

### 4.1. Practical Applications

A better technical–tactical performance was shown in the Late Maturity players in the modified competition. Basketball federations should revise the official rules for competitions with young players. In addition, federations should consider creating bio-banding competitions with the aim to generate a motivational and fair game.

### 4.2. Further Research

It is necessary to analyse the cognitive and motivational variables of these players. In addition, it is important to create similar studies at the girls' basketball level considering maturity timing and to observe the impact of these modifications in bio-banding competitions. Also, it is necessary to perform studies with a large sample size.

### 4.3. Limitations

The main limitation of this study was the short rest time between the formal and Modified Tournaments, and the players used masks and did not have assistants. It would

be important to use more teams of this level from different places to observe the general impact of this. Also, it was impossible for the players to practice previously with the modification of the height of the basket and the three- and four-point lines.

## 5. Conclusions

This study showed (a) that a reduced basket height and closer three-point line promoted a higher number of balls obtained, 1 vs. 1 situations, finished ball player actions, shots, and the efficacy of offence phases. (b) There was a significant increase in the number of balls obtained, 1 vs. 1 situations played, the number of plays finished with a lay-up or shot, number of received personal fouls, number of plays finished in 1 vs. 2, and those finished in equality and inferiority with a high defence opposition. (c) The modified version of the competition presented a higher number of technical–tactical actions in the Late Maturity players.

In summary, it is essential that sports federations organise specific competitions for beginners. Competitions with young players should be adapted to the players' biological development. However, scientific evidence should be used to support this decision-making process.

**Author Contributions:** Conceptualization, E.O.-T. and R.A.B.-A.; methodology, E.O.-T., R.A.B.-A. and J.M.G.-E.; software, E.O.-T., R.A.B.-A. and J.M.G.-E.; data curation, E.O.-T. and R.A.B.-A.; writing—original draft preparation, E.O.-T. and R.A.B.-A.; writing—review and editing, G.T.-L. and F.A.-L. All authors have read and agreed to the published version of the manuscript.

**Funding:** This research was funded by the Basket 2.0 project (No. 21076/PDC/19) granted by Fundación Séneca—Agencia de Ciencia y Tecnologia de la Región de Murcia and Consejo Superior de Deportes (19/UPB/21).

**Institutional Review Board Statement:** The study was approved by the Institutional Research Ethics Committee of University of Murcia (No. 2828/2020).

**Informed Consent Statement:** Informed consent was obtained from all subjects involved in the study. The study was approved by the Institutional Research Ethics Committee of the University of Murcia (No. 2828/2020).

**Data Availability Statement:** Data is contained within the article.

**Conflicts of Interest:** The authors declare no conflicts of interest.

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
