# Peer review of "Scaling Equipment Effect on Technical–Tactical Actions in U-13 Basketball Players: A Maturity Study"

_applsci, doi:10.3390/app14052193_

Round 1
Reviewer 1 Report
Comments and Suggestions for Authors
With regards to the employed statistical methodology, albeit seemingly correct, the authors ought to caution in their text that the inferences drawn should be interpreted with caution (this must be highlighted in the paper). This is attributed to the utilization of a small sample, comprising merely 37 observations. Concern arises from p-values closely approaching the threshold of 0.05, e.g. as evidenced in lines 136-137, with a p-value of 0.049. The question arises: is this effect genuinely significant or merely a consequence of the sample's limitation in capturing the true effect?
Comments on the Quality of English LanguageMust be improved.
Author Response
Dear reviwer
Thank you very much for your comments. We answer in attached document all your suggestions.
Thank you very much.
The authors

Reviewer 2 Report
Comments and Suggestions for Authors
-
Please proofread the paper again. E.g. Grammar error: “have modified and modifying only one rule, but have not offered specific proposals to “
-
How is the participant selected? Is it biased or unbiased? Does the participant know they are in the study? How will this bias the result?What is the health condition and competition physical and psychological readiness for these participants before the test?
-
The author may use better data visualization to present the data. The table is not so clear to show the difference?
-
How is the referee selected in the game? Is the referee biased?
-
Why do the authors choose 37 players? Can more players be selected to increase the credibility of the study?
Check typos and grammar errors.
Author Response
Dear reviewer
Thank you very much for your comments. We answer in attached document all your suggestions.
Thank you very much.
The authors

Reviewer 3 Report
Comments and Suggestions for Authors
The manuscript has a typical structure and follows good academic practice. The sections are clearly differentiated and allow easy navigation by the reader. On the other hand, it is necessary to improve certain sections of the manuscript in order to achieve the required academic refinement. In this context, it is necessary to do the following:
1. What is the purpose of work? The aim of the work stated in the Introduction section does not provide sufficient evidence as to why the study focuses on players up to 13 years of age (U13). Is the purpose of the study that only applies to this age group sufficient? It is necessary to provide more evidence on this and explain the context that supports this approach. For the stated reason, it is necessary to correct part of the manuscript at the end of the Introduction section.
2. What is the conclusion? It is necessary to introduce the Conclusion section and present the gloomy key results. Special emphasis should be placed on practical implications. That part is succinctly described in two sentences.
3. Does the research have any side effects in relation to the U13 population? First of all, we think of motivation and the development of basketball potential. Is it possible to get some guidelines for future research based on this?
4. Are these limitations the only limitations of the study? Are two tournaments in two days enough to observe? Is the age of the participants a limitation? What about younger basketball players? Does the research have a longitudinal component? What external factors could have influenced it? Please, honestly present any limitations of the study.
5. Some sentences should be reworded. For example "These competitions must be adapted to the psycho-evolutionary needs 356 of the participants." Certainly, we all must die, but everything else is not under must. This should be reworded according to good practice in the scientific papers.
Author Response

(The authors gave the same response as above.)

Reviewer 4 Report
Comments and Suggestions for Authors
Dear Author,
I would like to congratulate you for excellent work you have done.
Best regards,
Author Response

(The authors gave the same response as above.)

Round 2
Reviewer 2 Report
Comments and Suggestions for Authors
1. The response letter has several grammatical errors that make it hard to understand.
2. What is the recommended sample size to reach a convincing conclusion? Please clarify on point 5. It would be more convincing if the study involves more samples.
Author Response

(The authors gave the same response as above.)

Reviewer 3 Report
Comments and Suggestions for Authors
While the authors addressed the majority of the provided suggestions for revising the manuscript, they failed to address specific elements. Authors are required to do the following:
1. Limitations. In the previous review, the following was suggested to the authors: "Are these limitations the only limitations of the study? Are two tournaments in two days enough to observe? Is the age of the participants a limitation? What about younger basketball players? Does the research have a longitudinal component? What external factors could have influenced it? Please, honestly present any limitations of the study." The list of limitations should be expanded and a detailed description of the limitations should be added, including the guidelines from the previous review and the limitations implicit in “Further research”.
2. Academic writing. In the previous review, the following was suggested to the authors: "Some sentences should be reworded. For example "These competitions must be adapted to the psycho-evolutionary needs 356 of the participants." Certainly, we all must die, but everything else is not under must. This should be reworded according to good practice in the scientific papers." In the revised version of the text, there are still sentences containing "MUST". For example: "Basketball federations must revise the official rules for the young 359 competitors." Do they have to do that? Carefully revise the sentences in which this style of reasoning is present.
3. The text still contains a lot of spelling mistakes. For example: "... between adapted basketball and adunt basketball should be ..."; "creating a bio-banding competitions with theb propose to gertg a motivational" ... Proofreading is needed.
Comments on the Quality of English LanguageProofreading is needed.
Author Response

(The authors gave the same response as above.)
